

# Changes in beach shoreline due to sea level rise and waves under climate change scenarios: application to the Balearic Islands (Western Mediterranean).

Alejandra R. Enríquez[1], Marta Marcos[1], Amaya Álvarez-Ellacuría[2], Alejandro Orfila[1], Damià Gomis[1]

[1] IMEDEA (Universitat de les Illes Balears - CSIC), Esporles, Spain
[2] SOCIB, Balearic Islands Coastal Observing and Forecasting System, Spain

*Correspondence to*: Alejandra R. Enríquez (a.rodriguez@uib.es)

**Abstract.** In this work we assess the impacts in reshaping coastlines as a result of sea level rise and changes in wave climate. The methodology proposed combines the SWAN and SWASH wave models to resolve the wave processes from deep waters

up to the swash zone in two micro-tidal sandy beaches in Mallorca Island, Western Mediterranean. In a first step, the modelling approach is validated with observations from wave gauges and from the shoreline inferred from video monitoring stations, showing a good agreement between them. Afterwards, the modelling setup is applied to the 21st century sea level and wave projections under two different climate scenarios, RCP45 and RCP85. Sea level projections were retrieved from state of the art regional estimates, while wave projections were obtained from regional climate models. Changes in the coastline are

explored under mean and extreme wave conditions. Our results indicate that the studied beaches would suffer a coastal retreat between 7 and up to 50 m, equivalent to half of the present-day aerial beach surface, under the climate scenarios considered.

## 1 Introduction

Rising sea levels represent one of the major threats for coastal regions, causing submersion, erosion and increased vulnerability to extreme marine events, among other negative impacts (Nicholls and Cazenave, 2010). It is expected that such effects will

be aggravated in the coming decades as sea level rise accelerates in response to global warming (Church et al., 2013) and coastal population and development grow (Hanson et al., 2011).

Several studies have related coastline retreat during the last decades with sea level rise (e.g. Feagin et al., 2005; FitzGerald et al., 2008), although other relevant processes have also been identified. These include oceanic forcing by wave climate and storms, direct or indirect human actions (e.g. mining activities or fluid extraction) and local features such as coastal morphology

(Cazenave and Le Cozannet, 2014). Coastline retreat has important environmental impacts, but also socio-economic implications as it affects population, infrastructures and assets. The impact of sea level rise in the shoreline position has therefore become a subject of increasing concern, particularly in densely populated regions with high urban development. This is the case of many Mediterranean regions, whose economy, which constitutes about 14% of the total Gross Domestic Product of the EU (Eurostat, 2011), largely relies on tourism based on beach and other seaside recreational activities. Thus, sea level



rise and its potential impacts are key factors that must be incorporated in coastal risk management and climate change adaptation measures.

In this paper we investigate the shoreline changes in two anthropized micro-tidal sandy beaches located in Mallorca (Balearic Islands, Western Mediterranean Sea). The potential impacts of a shoreline retreat would increase the vulnerability of the near-

shore infrastructures. In addition, both are typical tourism-oriented beaches in urban environments of the Mediterranean region, so that their reduction or disappearance would be detrimental for the local economies.

The impact of sea level rise along sandy coastlines consists of two processes, namely inundation and erosion. Increased sea levels allow waves and surges to act at higher levels landward in the beach profile, increasing erosion rates (Zhang et al., 2004). However, in this study the beach erosion has not been considered, which means that our estimates of landward migration

of the coastline could be biased low if erosion rates increase and sediments are carried offshore. This assumption is further discussed later. Some earlier studies have explored the potential impact of future sea level rise on shoreline changes, although without taking into account changes in the wave climate (see e.g. Wu et al., 2002; Stive, 2004; Poulter and Halpin, 2008; Le Cozannet et al., 2014). Others have addressed the impact of waves, including extreme events, erosion rates, morphological changes, flooding, and vulnerability of infrastructures but sometimes without including changes in sea level (see e.g. Ruju et

al., 2012; Vera Guimarães et al., 2015; Medellin et al., 2016). Here we combine both, regional sea level and wave projections in the Western Mediterranean under two different climate scenarios for 2100, in line with works as Villatoro et al (2014). For sea level we have not used global projections; instead, we have estimated regional values for each of the different contributors to sea level rise using state-of-the-art results. For the wave regime, we have combined the SWAN and SWASH numerical models to resolve the wave processes from offshore wave conditions up to the beach. The offshore conditions have been

obtained from regional wave projections for the Mediterranean Sea and we explored changes in the mean regime as well as in the extreme waves. In summary, we have used the most up-to-date ocean forcing to represent projected changes in the oceanic conditions under climate change scenarios. Our results would thus be consistent with adopted values of sea level rise and the definition of the future climate scenarios (Brunel and Sabatier, 2009; Tamisiea and Mitrovica, 2011; Church et al., 2011; IPCC, 2013).

The paper is organized as follows. Section 2 is devoted to the description of the study areas, the characteristics of the wave climate, the data available and the numerical approach. The validation of the methodology, which includes the comparison between modelled and observed shallow water waves and coastline positions, is presented in Section 3. Section 4 describes the shoreline changes obtained under different climate change scenarios. Finally, a summary and some conclusions are presented in Section 5.

**2 Data and Methods**

Cala Millor and Playa de Palma are two micro-tidal sandy beaches located in Mallorca Island (Balearic Islands, Western Mediterranean Sea, Figure 1). Cala Millor is 1.7 Km along-shore by 35-40 m cross-shore and is exposed to offshore waves



from NE to ESE. The wave regime in deep waters has a significant wave height (Hs) of 1 m and a peak period (Tp) of 4 s. Playa de Palma beach is 4 km along-shore by 30-50 m cross-shore and is exposed to offshore waves conditions from SE to SW, with a Hs of 0.7 m and a Tp of 4.8 s. Figure 2 characterizes the mean wave regime offshore both sites using self-organizing maps (SOM) that have been built with a 58-yr wave hindcast (see section 2.5 for more details). SOM display graphically the temporal distribution of Hs, Tp and wave direction (in arrows). The results evidence that low-energy states are dominant at both sites and that, overall, Cala Millor is more energetic than Playa de Palma.

Playa de Palma and Cala Millor beaches are part of the beach monitoring programme of the Balearic Islands Coastal Observatory and Forecasting System (SOCIB) since 2011 (Tintoré et al., 2013). This programme includes periodic topography and bathymetry surveys, continuous video-monitoring of the shoreline position and in-situ measurements of near-shore waves and currents, among others. In addition, a dedicated field survey (RISKBEACH) was undertaken in Cala Millor in March-April 2014, during which higher resolution observations were obtained (Morales et al., 2016). Specific data used in the present work are described in the following.

### 2.1 Topo-bathymetric surveys

Bathymetry surveys were conducted using a single-beam echo-sounder "BioSonics DT/DE Series Digital Ecosounder" in Cala Millor beach and a multi-beam echo-sounder "R2Sonic2020" in Playa de Palma beach. The final spatial resolution was 1 m cross-shore and 2 m alongshore in Cala Millor and 0.5 m x 0.5 m in Playa de Palma. These measurements were complemented with topographies of the aerial beach obtained using a survey grade RTK-GPS (Real Time Kinematic – Global Position System) mounted in a backpack carried by a human walker. These detailed beach topo-bathymetries were surveyed under calm conditions.

### 2.2 Hydrodynamic data

In Cala Millor, nearshore hydrodynamic data were obtained from three directional wave Acoustic Waves and Currents (AWAC) sensors located at 8 m, 12 m and 25 m water depths; the AWACs were deployed as part of the RISKBEACH field survey, which covered from 12-March-2014 to 14-April-2014. Offshore hourly hydrodynamic data were recovered from Capdepera buoy, located 36.45 km northeast of Cala Millor at 48 m depth (see Figure 1 for location). The buoy has been operative during the period 1989-2014 as part of Puertos del Estado (the Spanish Holding of Harbours) buoys network. On the other hand, in Playa de Palma, wave data come from a coastal buoy located at 23 m depth and an ADCP deployed at 17 m depth, both operating since January 2012 as part of the SOCIB beach monitoring programme.

### 2.3 Video imagery data

Five and fourteen cameras images were used to measure the coastline position along Cala Millor and Playa de Palma beaches, respectively. These cameras are part of the video-based coastal zone monitoring system called SIRENA developed by SOCIB and IMEDEA. Departing from images taken at 7.5 Hz the SIRENA system generates statistical products that after specific





post processing provide quantitative information of hydrodynamics and morphodinamics (Nieto et al., 2010). Specifically, the coastline is obtained from the timex image consisting in the addition of all images captured during 10 minutes (a total of 4500 images) and applying a post-processing of cluster classification. After applying different corrections to overcome the coarser resolution of the far field camera images as well as rectifying the perspective projection, the coastline is georeferenced in a

world coordinate system.

## 2.4 Numerical approach

With the aim of simulating the shoreline changes under given offshore conditions, the SWAN (Booij et al., 1999) and SWASH (Zijlema et al., 2011) models are combined to resolve the wave processes from deep waters up to the swash zone. SWAN is a third generation wave model that solves the spectral action balance equation for the propagation of wave spectra

(http://swanmodel.sourceforge.net/). This model allows an accurate and computationally feasible simulation of waves in relatively large areas. On the other hand, SWASH is a phase resolving non-hydrostatic model governed by the nonlinear shallow-water equations with the addition of a vertical momentum equation and non-hydrostatic pressure in horizontal momentum equations (http://swash.sourceforge.net/). Due to its computational cost, the application of SWASH is restricted to small areas. The combination of both models allows high resolution and accurate results with less computational cost.

For the present study, SWAN simulations were performed in a stationary mode over two regular nested grids. In Cala Millor, the coarser grid covers a domain of 21 km x 21 km with its lowest left vertex at 39.53ºN, 3.38ºE (Figure 3) and a resolution of 149 m x 119 m in the x and y directions, respectively. The size of the finer grid is 9.5 km x 9.5 km with its lowest left vertex at 39.6ºN, 3.38ºE and a resolution of 60 m. The coarse grid in Playa de Palma beach covers a domain of 21.5 km x 27.7 km, with its lowest left vertex at 39.31ºN, 2.5ºE (Figure 4) and a resolution of 100 m x 100 m in x and y directions. The domain of

the finer grid is 13 km x 10.8 km starting at 39.47ºN, 2.58ºE, with a resolution of 50 m x 50 m. In all cases, the SWAN output consisted of the 2D variance energy density spectrum and the spectral parameters of propagated wave conditions. Each output SWAN spectra corresponded to one hour of simulation and were used as the input wave conditions of SWASH.

SWASH simulations in Cala Millor were performed on a 1.5 km x 3.2 km rectangular grid with its lowest left vertex at 39.57ºN, 3.38ºE and a resolution of 3 m x 3 m (Figure 3), with a maximum depth at 17 m. A larger SWASH domain was required in

Playa de Palma: we used a 3 m x 3 m grid covering a domain of 3 km x 7 km starting at 39.47ºN, 2.75ºE and tilted 45º in order to orient the wave maker boundary parallel to the beach, at 15 meters depth. The SWASH simulations lasted for 30 min, with a time step of 0.05 s to keep the Courant number between 0.01 and 0.5. The initial wave conditions imposed at the eastern boundary in Cala Millor and at the southwestern boundary in Playa de Palma, corresponded to the 2D variance energy density spectrum field provided by the corresponding SWAN simulations. The final output consisted of instantaneous water level

elevations in the whole domain and the position of the coastline at each time step.





## 2.5 Forcing of numerical models

The SWAN-SWASH model setup was run under present-day and future climate conditions in both domains. In order to validate model performance, the present-day runs, forced with realistic offshore waves, were compared against measured nearshore wave parameters. Deep water conditions were retrieved from the SIMAR database (Pilar et al., 2008), a 58-years wave re-

analysis generated with the WAM model (WAMDI GROUP, 1988). The re-analysis, which is freely distributed by Puertos del Estado, covers the Western Mediterranean and provides 3-hourly wave data up to 2011 and hourly data since then. The two closest SIMAR grid point to each of the domains were selected to force the SWAN model for the periods of validation (as detailed later). Although this data set has already been evaluated against observations (Pilar et al., 2008; Martinez-Asensio et al., 2013, 2015), we further compared one of this output with the offshore waves observed at Capdepera buoy in order to ensure

the reliability of the forcing in our particular periods and locations (section 3.1).

Projected sea level rise together with changes in the wave climate define the future conditions under which the models are run to evaluate the shoreline changes. A summary of the values used for sea level and waves is presented in Table 1. Sea level projections by 2100 were computed following Slangen et al (2014), who provided the regional distribution of the different contributors to sea level change under two climate change scenarios, namely Representative Concentration Pathway (RCP) 45

and RCP85 (Moss et al., 2010), which are representative of moderate and large emission scenarios, respectively. Slangen et al (2014) used an ensemble of 21 Atmosphere-Ocean coupled General Circulation Models (AOGCMs) from the Coupled Model Intercomparison Project Phase 5 (CMIP5) archive, to estimate changes in ocean circulation and heat uptake contribution, atmospheric loading, land ice contribution (including all glaciers, ice caps and ice sheets on Greenland and Antarctica), groundwater depletion and mass load redistribution worldwide, together with the associated uncertainties for each term. As

the regional distribution of each component was provided, we selected the Mediterranean region and averaged the sum of the components. Our results lead to a regional sea level rise of 48±23 cm and 67±31 cm by 2100 for RCP45 and RCP85, respectively. Uncertainties correspond to 1−σ deviation from the ensemble mean (Slangen et al., 2014).

On the other hand, changes in the wave climate during the 21$^{st}$ century were obtained from regional wave projections over the Western Mediterranean (Puertos del Estado et al., 2016). These projections were carried out using the WAM model with a

spatial resolution of 1/6º (over the same grid as the SIMAR data base) and forced with a set of dynamically-downscaled surface wind fields from AOGCMs. A total of 6 simulations were considered, five corresponding to the A1B scenario and one to the A2 scenarios (IPCC SRES 2000). Each projection was accompanied by a control simulation representing the climate of the last four decades of the 20$^{th}$ century, as it is usual practice. As the regional wave projections were computed before the adoption of the new set of RCP scenarios, for our purposes we assume here that A1B (A2) scenario is equivalent to RCP45 (RCP85).

Changes in the mean and extreme wave regimes were assessed by computing the differences between the values averaged over the period 2080-2100 (from the future projections) and those averaged over 1980-2000 (from the control simulations). The obtained differences were then added to the hindcasted values from the re-analysis, which represent our best approach to the actual present-day climate. For each beach, the closest grid points (the same location as for the SIMAR database) were selected



to simulate the future wave climate. At the point representative of deep water wave regime of Cala Millor, the mean Hs is 1.20 m and 0.95 m for the A2 and A1B scenarios, respectively, while in Playa de Palma the values are 0.63 m and 0.65, respectively. The storm events were assessed by computing the 10-years return periods by fitting a Generalized Pareto Distribution to each time series. The values obtained were 4.5 m and 4.2 m under A2 and A1B climate change scenarios in Cala Millor and 4.3 m

and 4.4 m in Playa de Palma. Given the similarities between the two wave climate change scenarios, we finally use a single (mean) value for the simulations (see Table 1). Regarding to the wave direction the changes are negligible and remain unchanged in the future simulations.

In summary, six wave cases were carried out for each site to predict the shoreline changes under mean conditions: one for each of the two sea level rise scenarios (RCP45 and RCP85) and one for their respective upper and lower uncertainty limits (i.e.

plus 1-σ and minus 1-σ). In addition, four simulations were performed for extreme conditions: due to computational constraints we focused on the worst case scenarios, that is, the occurrence of the 10-years return level storm occurring over the two sea level scenarios and their upper limit (i.e., for the mean value and the mean value +1-σ).

## 3. Evaluation of model setup under present-day climate conditions

### 3.1 Comparison with wave observations

As described above, the SIMAR wave re-analysis has been taken as representative of the offshore wave conditions and used to force the numerical model setup. To illustrate its reliability, the time series at the closest grid point in Cala Millor was compared against observations from the nearby Capdepera buoy. The time series and scatter plots of the measured and modelled statistical wave parameters (Hs, Tp, θ) are shown in Figures 5 and 6 for a 3-months period (January-March 2014). The root mean square error (RMSE) and the correlation coefficient (ρ) between observed and modelled parameters are quoted

in the figures. Results show that the hindcast agrees well with the observed Hs and Tp with correlations over 0.8 and small RMSE. For wave direction, however, the correlation decreases down to 0.5, mostly due to the fact that the WAM resolution cannot properly resolve the coastal topography near the SIMAR location. A closer look at Figure 5 (bottom panel) reveals that SIMAR gives waves from NW which are not recorded by the buoy. However, waves from the dominant directions (i.e. from N to SE) are not affected and, therefore, the Hs and Tp have enough accuracy to represent the wave climate of this offshore

area.

Despite the differences found in the wave direction, the advantages of using re-analysed data instead of observations for the input wave in SWAN are evident: first, the modelled time series are complete, while observations are often gappy; and second, the deep water waves can be propagated over large domains thus providing values close to our two areas of study. Although our validation of the numerical hindcast is limited to a single grid point close to Cala Millor, we rely on previous assessments

(e.g. Martínez-Asensio et al., 2013) and assume that the re-analysis is equally valid for Playa de Palma.

The output of the SWAN model was also validated against observations in the two beaches. In Cala Millor the results of SWAN forced with SIMAR data were compared with nearshore wave observations during the period from 14- March to 14-





April-2014 (i.e. a total of 755 hours of simulation). The closest grid points of the SWAN model to each of the three directional wave ADCP were selected. Resulting correlations, RMSE and biases are listed in Table 2 for the three ADCP and for the three wave parameters. Overall, the statistical parameters show good agreement between measurements and the model output, with correlations of 0.9 for Hs and Tp and over 0.7 for the wave direction. To further illustrate the model performance, observed

and modelled time series are plotted in Figures 7 and 8. Both reflect the ability of the model to capture the magnitude and variability of nearshore waves. Nevertheless, during the storm events recorded (as in March 28th), the model underestimates the observed Hs by up to 30 cm.

In Playa de Palma, the simulated waves were compared with the observations from a buoy moored at 23 meters depth and with an ADCP at 17 m depth for the period of 1-September to 30-September 2015 (i.e. a total of 720 hours of simulation). The

results are summarized in Table 3 and the time series are plotted in Figures 9 and 10. Like in Cala Millor, there is a good agreement in Hs with correlations over 0.9. For Tp, however, observations display higher variability than modelled data, which makes the correlations to drop to 0.3 and the bias to reach 0.6 s (see Figure 10). The differences between observed and modelled wave directions are also larger than in Cala Millor, with non-significant correlations. The reason for the discrepancies in wave direction is probably the inability of the model to accurately represent the wave diffraction occurring at the SE of the bay of

Palma, where the buoy and the ADCP are located. This area is protected by a headland (see Figure 4) that may cause worse results in wave direction.

## 3.2 Comparison with observed shoreline position

A total of four and three simulations were carried out with the SWASH model for Cala Millor and Playa de Palma beaches, respectively, in order to validate the model results with measurements of shoreline positions. The dates chosen for the

validation correspond to dates in which the video monitoring provided good quality images being also close to the dates were the bathymetry surveys were performed (they are listed in Tables 4 and 5). Wave makers were defined at the eastern boundary of the SWASH model domain in Cala Millor and at the south-western boundary in Playa de Palma, in both cases with the SWAN wave conditions. These input wave conditions for the validation process are specified in Tables 4 and 5 for the indicated dates.

Observed and modelled shoreline changes for each case study are compared in Figures 11 and 12 along the two beaches. Results show that the modelled shorelines line up with observations in all cases. In Cala Millor the agreement is better in the central part of the beach, while some differences are found in the northern and southernmost sector. It is important to remark that images obtained from the beach cameras are increasingly uncertain with the distance from the cameras. Therefore, part of the difference between measured and simulated shoreline at the ends may come from this error in measurements. In Playa de

Palma only the area between 39.51 º N and 39.53 º N is used for the comparison as this is the stretch of the shoreline where the video-system has the requested quality. We will also restrict to this sector the discussion on future projections.

In addition to the figures, the RMSE and biases between observations and model results have been calculated for each case and are listed in Tables 4 and 5. These statistics must be set in a proper context in order to evaluate how good the model





performance is. To do so, we have estimated the temporal variability of the shoreline position as the standard deviation (cross-shore) at each along-shore position for which we have used 10 coastlines measured from video monitoring. In Cala Millor we observe higher variability, calculated between April and May 2014, in the central part of the beach (mean value of 8.4 m) and lower towards the ends, with a mean value along the entire beach of 5.5 m. Figure 13 shows the shorelines simulated for the

case studies (red lines), the corresponding measured shorelines (blue lines) and the variability of the shoreline (grey area), zoomed around an area at the centre of the beach. In the case of Playa de Palma, the shoreline displays a cross-shore variability of 6 m in the area arouns the centre of the beach and lower at the extremes, with a mean value of 3 m, as calculated with observations between August and October 2014. The results are plotted in Figure 14 in which again the central area has been zoomed in order to highlight the differences. Notably, the modelled shorelines are very similar to each other, because the

forcing was also similar in the three case studies.

## 4. Shoreline changes under climate change scenarios

Since the model performance for present-day climate conditions is considered to be satisfactory, the same model setup will be used to assess the response of the shoreline under future climate change scenarios. Shoreline changes were simulated for both, mean conditions and extreme waves (the latter being defined here as Hs corresponding to the 10-year return level) for the

RCP45 and RCP85 climate change scenarios.

Future projected changes in shoreline are evaluated assuming that the present-day beach profile remains constant. In order to check the limitations of this assumption we have run a numerical one-dimensional model capable of estimating profile changes under different mean sea level conditions. The model used here is PETRA (Gonzalez et al., 2007) and it was run for the central profile of each beach. The simulations were forced with mean waves over two mean sea level cases, namely present-day sea

level and the worst case scenario (i.e. RCP85 + 1-σ corresponding to 98 cm). The model outputs indicate that in these environments profile change due to sea level rise are negligible (of the order of sandbar formation and mostly eroding the berm) and therefore we will not take it into account in this work.

The present-day modelled coastline was used as a reference to assess the changes under climate change scenarios. The loss of aerial beach, defined here as the landward migration averaged over the entire beach, is indicated in Tables 6 and 7 for each

simulation and for mean and extreme conditions expected under climate change scenarios. For the extreme conditions, also the maximum loss is listed. In addition, Figures 15 and 16 illustrate the maximum change in the shoreline position obtained for Cala Millor and Playa de Palma (corresponding to extreme wave conditions under the RCP85 + 1-σ scenario). Major changes are projected to occur in the central part of Cala Millor beach, where it shows the higher variability (see Figure 13). Larger relative impacts (loss of width), however, are projected towards the extremes of the beach, as these are the narrower

sectors. In Playa de Palma, the projected changes in the shoreline are quite uniform along the beach.

Since projected changes in Hs by 2100 are small, their potentially hazardous effects depend primarily on the mean sea level with which they are combined. In Cala Millor, the averaged coastline retreats ranges between 7 m under moderate/low scenario



and 24 m with the highest sea level rise considered. During extreme wave conditions the shoreline would retreat up to 29 m on average and may reach 49 m at some parts of the beach. With such values the flooding would reach the urbanized area over the promenade. However, it must be pointed out that the topography does not include the height of the wall backing the beach, so that the flooding extension could actually be overestimated. In Playa de Palma the average coastline retreat ranges from the

7 m obtained for the low scenario to the 21 m obtained for the worst case scenario. Under extreme conditions, the loss of Playa de Palma beach increases with higher sea level rise and, in all the cases investigated, the water level reach the promenade at least in part of the domain (Table 6).

## 5. Summary and conclusions

In this paper we have investigated the capabilities of state of the art numerical models to reproduce the changes in the shoreline

position in Cala Millor and Playa de Palma beaches. These two case studies were selected for two main reasons. First, they are representative of many other anthropized beaches in the Balearic Islands (and of many other beaches of the Mediterranean Sea): they are beaches located in urbanized areas, backed by walls and therefore with limited possible landward migration of the shoreline. Second, these two sites are part of the beach monitoring programme carried out by SOCIB and, consequently, a wide and complete set of observations is available allowing the validation of the numerical models against measurements.

Furthermore, the two beaches are exposed to offshore wave conditions from different directions and different wave heights, with Playa de Palma being located inside a bay and Cala Millor facing the open sea.

Much effort has been devoted to the validation of the model set up to ensure that the chosen combination of SWAN – SWASH models is able to reproduce the shoreline variability within a reasonable accuracy. In both cases, modelled and observed Hs from near-shore instruments are in very good agreement, with correlations over 0.9. This increases our confidence in the

forcing of the SWASH model. In turn, a satisfactory correspondence between observed and modelled shoreline position has been found. The agreement between modelled and observed shorelines is better in the central sector of the beaches. This is because the observations derived from the video monitoring system are more reliable close to the location of the cameras and also because the SWASH model configuration requires a smooth bathymetry which can misrepresent some parts of the shore, as is the case of the southernmost sector of Cala Millor where a bed rock and a small cliff distort the wave field.

Regarding the projections of the shoreline changes under climate scenarios of sea level and wave climate, a major assumption of our study is that the morphology of the beach will not change in the future. That is, both the beach shape and the profile will be the same under the climate conditions at the end of the century. It is well known that beach profile evolves in response to storms, moderate wave conditions and sea level rise causing changes in the beach morphology (e.g. erosion followed by recovery episodes, see Short et al., 1996; De Falco et al., 2014; Smallegan et al., 2016). The justification of constant beach

shape is largely justified by the fact that there are no significant changes in the energy flux in wave projections that may force a change in the shape due to oceanic forcing. Furthermore, we have also demonstrated that the changes in the beach profile



play a minor role in the shoreline retreat due to sea level rise and waves. On top of the above reasons, we can hardly avoid the simplifications, as numerical approaches reproducing the long term morphological response of the beach do not exist so far.

Under the assumptions outlined above we have found that the retreat in the future shoreline in both sites, Cala Millor and Playa de Palma are primarily a consequence of waves acting onto a higher mean sea level. It must be remarked that changes in the wave climate are small and the impact of extreme waves increases mostly because they are projected to occur concurrently with higher sea levels. Our results indicate that the beach regression varies between 7 and 24 m along Cala Millor and between 7 and 21 m in Playa de Palma, depending on the climate change scenario considered. This lost is further exacerbated under moderate (return period of 10 years) storm conditions, which may induce a temporary flooding reaching over 49 m in Cala Millor and 30 m in Playa de Palma, thus likely overtopping the walls of the promenade. The Playa de Palma coastal retreat is lower than in Cala Millor due to the steeper slope of the beach profile. As pointed out above in the introduction, our approach does not consider beach erosion, which means that the above estimates are conservative and could be biased low if erosion acts removing beach sediments and accelerating aerial beach loss (Brunel and Sabatier, 2009).

Playa de Palma and Cala Millor, like many other typical urban Mediterranean beaches, are subject to high touristic pressure, especially during the summer season, and thus concentrate valuable assets and infrastructures. Since tourism constitutes the main economic activity of a large fraction of the region, the social, environmental and economic impacts of future sea level rise are anticipated if no adaptation measures are implemented.

**Acknowledgements**

This work is supported by the CLIMPACT (CGL2014-54246-C2-1-R) funded by the Spanish Ministry of Economy) and MORFINTRA (CTM2015-66225-C2-2-P). A. R. Enríquez acknowledges an FPI grant associated with the CLIMPACT project. M. Marcos acknowledges a "Ramón y Cajal" contract funded by the Spanish Government. We thank Puertos del Estado for providing deep water wave data from the SIMAR database. Topo-bathymetries and video-monitoring observations are part of the beach monitoring facility of SOCIB. The authors are grateful to Dr. A. Slangen for providing the data for the regional sea level rise scenarios.

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

**Table 1. Input condition of the model setup under climate change scenarios for the two beaches. See the text for details on their computation.**

|  | Climate scenario | **Cala Millor** | **Playa de Palma** |
|---|---|---|---|
| Sea level rise (mean±1σ, in cm) | RCP45 | 48±23 | |
| | RCP85 | 67±31 | |
| Hs | A1B | 1.1 | 0.64 |
| | A2 | | |
| Hs 10-year return period, in m | A1B | 4.5 | 4.4 |
| | A2 | | |

30





**Table 2: Comparison between SWAN results and nearshore wave observations in Cala Millor beach. The period spanned by the series is from 14- March to 14-April-2014.**

|  | ADCP 8 m | | | ADCP 12 m | | | ADCP 25 m | | |
|---|---|---|---|---|---|---|---|---|---|
|  | RMSE | BIAS | Corr. | RMSE | BIAS | Corr. | RMSE | BIAS | Corr. |
| $H_s$ (m) | 0.13 | 0.01 | 0.97 | 0.18 | 0.03 | 0.95 | 0.23 | 0.11 | 0.95 |
| $T_p$ (s) | 1.21 | 0.02 | 0.94 | 1.24 | 0.01 | 0.94 | 1.22 | -0.22 | 0.93 |
| $\theta_p$ (º) | 25.60 | 6.30 | 0.74 | 29.20 | 3.65 | 0.80 | 40.43 | 14.20 | 0.72 |

**Table 3: Comparison between SWAN results and nearshore wave observations in Playa de Palma beach. The period spanned by the series is from 01 to 30 September-2015.**

|  | Buoy 23 m | | | ADCP 17 m | | |
|---|---|---|---|---|---|---|
|  | RMSE | BIAS | Corr. | RMSE | BIAS | Corr. |
| $H_s$ (m) | 0.19 | 0.12 | 0.95 | 0.17 | 0.09 | 0.95 |
| $T_p$ (s) | 1.56 | 0.40 | 0.32 | 1.74 | 0.60 | 0.34 |
| $\theta_p$(º) | 46.19 | 18.5 | 0.29 | 49.4 | 30.9 | NS |

**Table 4. Dates and forcing conditions of the SWASH simulations and results of the validation against observed shoreline position in Cala Millor beach.**

|  | $H_s$ (m) | $T_p$ (s) | $\theta_p$(º) | RMSE (m) | BIAS (m) |
|---|---|---|---|---|---|
| 27-March | 1.6 | 8.3 | 13 | 5.7 | 3.2 |
| 28-March | 0.8 | 7.9 | 28 | 2.7 | -0.6 |
| 1- April | 0.5 | 5.5 | 137 | 6.5 | -3.2 |
| 2- April | 1.1 | 5.7 | 134 | 5.4 | -3.2 |

15 **Table 5. Dates and forcing conditions of the SWASH simulations and results of the validation against observed shoreline position in Playa de Palma beach**

|  | $H_s$ (m) | $T_p$ (s) | $\theta_p$(º) | RMSE (m) | BIAS (s) |
|---|---|---|---|---|---|
| 03- Sept. | 0.4 | 3.7 | 154 | 6.1 | 1.5 |
| 15- Sept. | 0.6 | 6.7 | 223 | 5.9 | 1.7 |





| **28- Sept.** | 0.4 | 2.7 | 47 | 5.8 | 1.4 |
|---|---|---|---|---|---|

**Table 6: Loss of aerial beach (defined here as the landward migration of the shoreline averaged over the entire beach) for both, the mean and extreme conditions expected under climate change scenarios in Cala Millor (in m). For the extreme conditions, also the maximum loss is quoted.**

| *Sea Level Rise (climate scenario ± uncertainty, in cm)* | **Mean conditions** | **Extreme conditions** | |
|---|---|---|---|
| | **Mean loss (m)** | **Mean loss (m)** | **Max loss (m)** |
| *0.25 (RCP45 -1σ)* | 7.2 | - | - |
| *0.36 (RCP85 -1σ)* | 10.7 | - | - |
| *0.48 (RCP45)* | 11.7 | 18.5 | 29.4 |
| *0.67 (RCP85)* | 17.5 | 21.8 | 38.0 |
| *0.71 (RCP45 +1σ)* | 17.5 | 24.6 | 39.5 |
| *0.98 (RCP85 +1σ)* | 24.2 | 29.0 | 49.3 |

**Table 7. Loss of aerial beach (defined here as the landward migration of the shoreline averaged over the entire beach) for both, the mean and extreme conditions expected under climate change scenarios in Playa de Palma (in m). For the extreme conditions, also the maximum loss is quoted.**

| *Sea Level Rise (climate scenario ± uncertainty, in cm)* | **Mean conditions** | **Extreme conditions** | |
|---|---|---|---|
| | **Mean loss (m)** | **Mean loss (m)** | **Max loss (m)** |
| *0.25 (RCP45 -1σ)* | 7 | - | - |
| *0.36 (RCP85 -1σ)* | 8.2 | - | - |
| *0.48 (RCP45)* | 11.3 | 17 | 30 |
| *0.67 (RCP85)* | 14.8 | 20.5 | 30 |
| *0.71 (RCP45 +1σ)* | 15.7 | 23.4 | 30 |
| *0.98 (RCP85 +1σ)* | 21.4 | 27.9 | 30 |





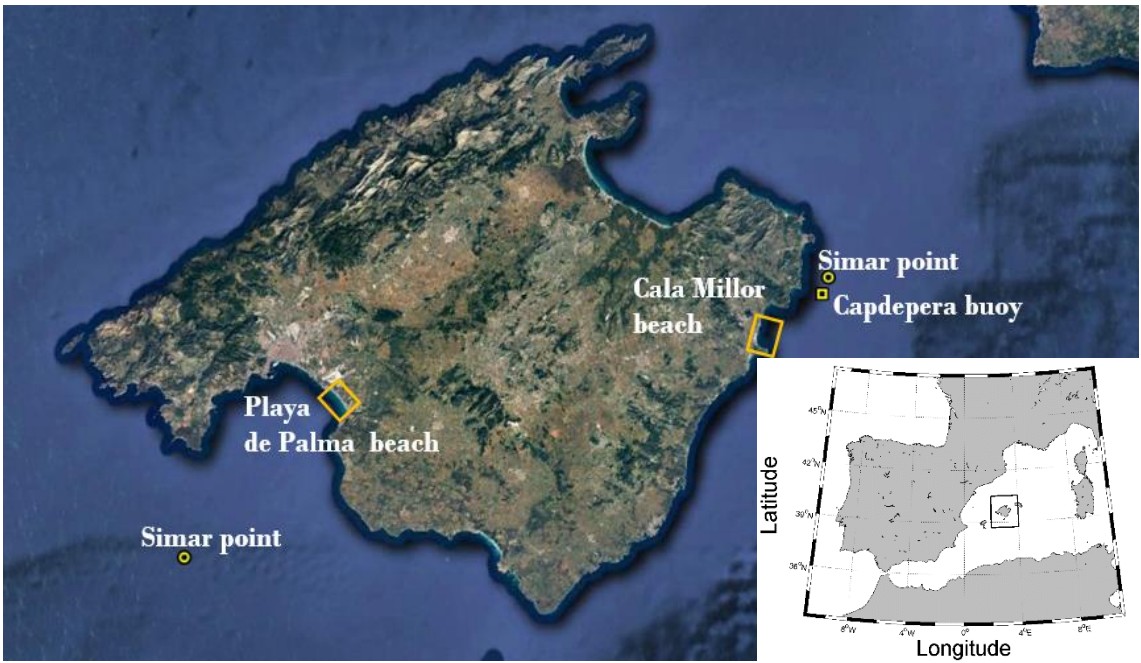

**Figure 1: Mallorca Island with Cala Millor and Playa de Palma beaches marked with orange squares. SIMAR grid points used to characterize the offshore wave climate and the Capdepera wave buoy are also marked. The inset map represents the Western Mediterranean basin.**

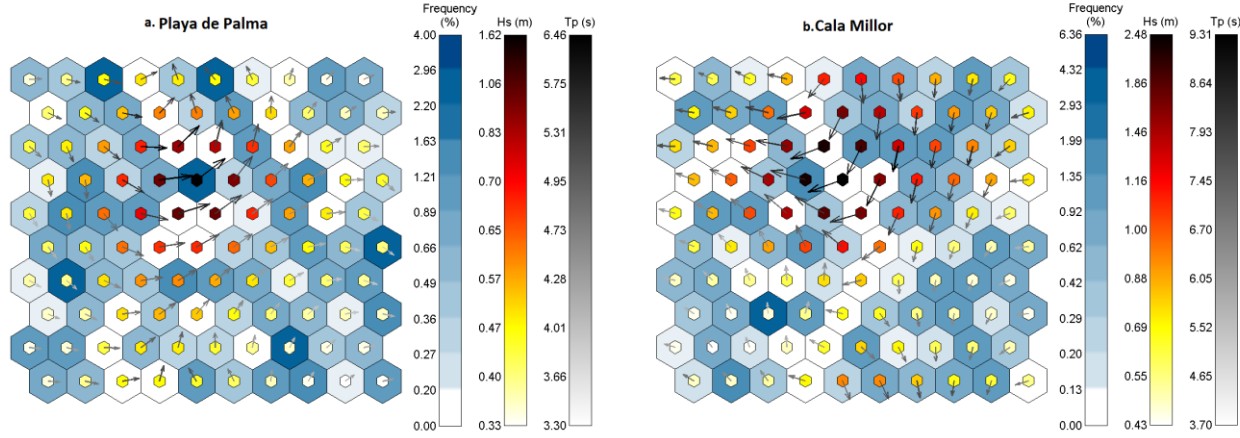

**Figure. 2: Playa de Palma and Cala Millor self-organizing maps (SOM). SIMAR databases are shown in 100 cells displaying the more representative deep water sea conditions at Playa de Palma (a) and Cala Millor (b) beaches. The blue colour illustrates the frequency of the sea states, together with the Hs in meters (yellow to red), the period in seconds (white to black) and the direction in arrows. It can be seen that the more energetic conditions come from the SW in Playa de Palma and from the NE in Cala Millor,**



**also the more frequency waves are low energy in both sites.**

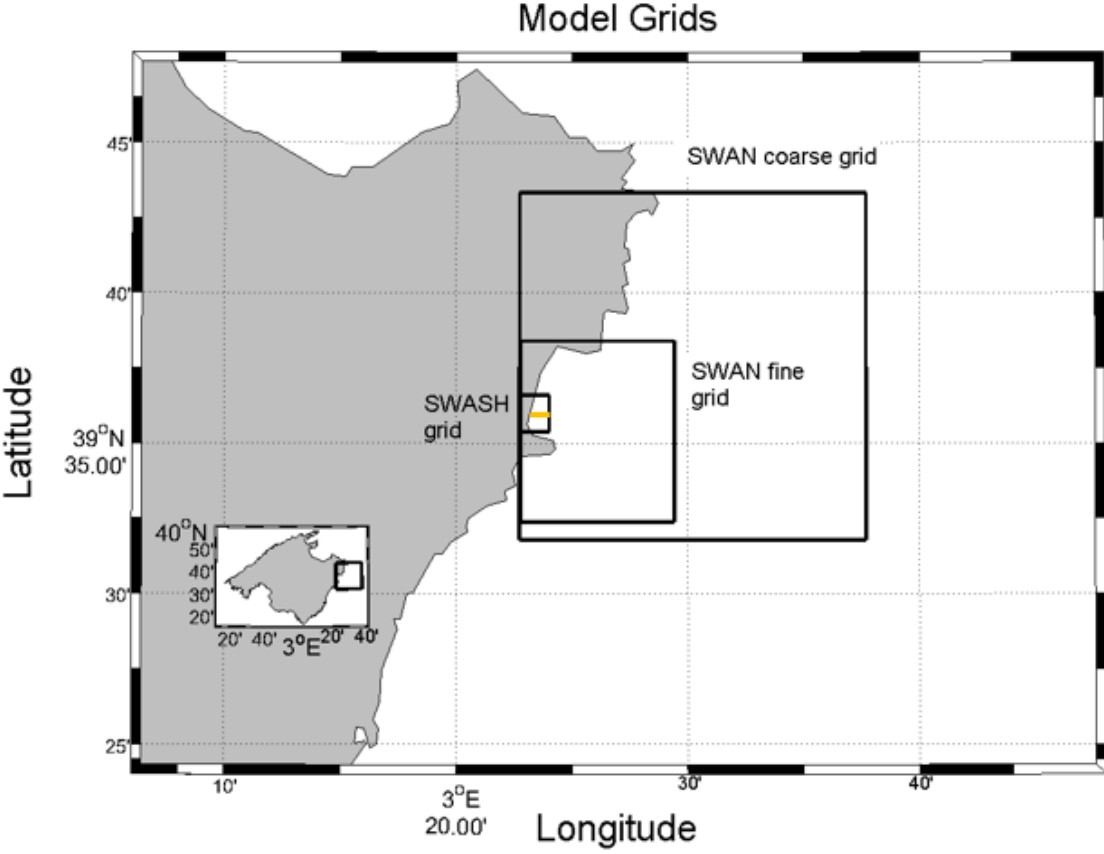

**Figure. 3: SWAN and SWASH computational domains for Cala Millor beach. Yellow line indicates the sector where the three ADCPs are located.**




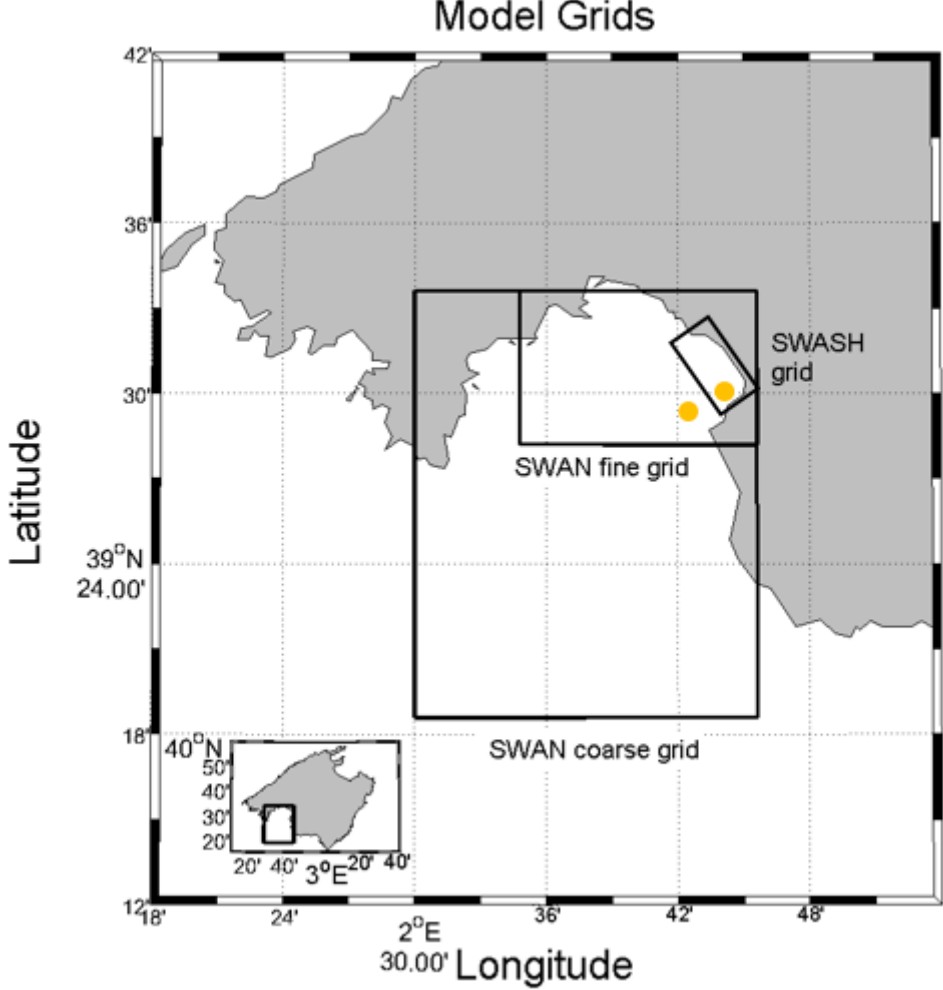

**Figure 4: SWAN and SWASH computational domains for Playa de Palma beach. Yellow dots indicate the locations of the shallow water wave buoy and ADCP.**




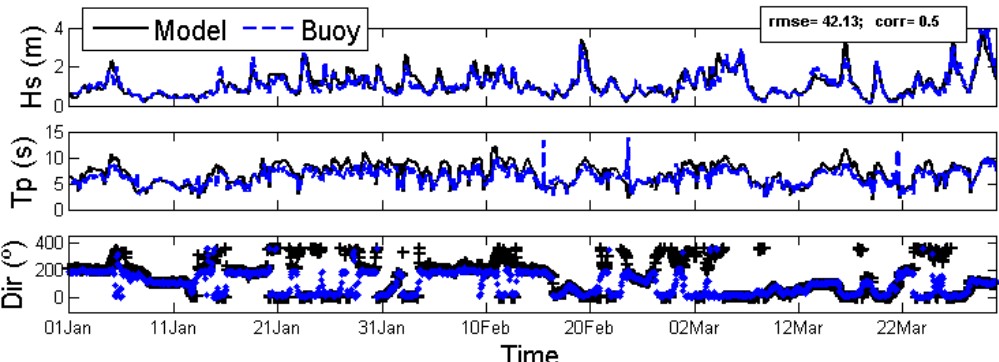

**Figure 5 Capdepera buoy observations (blue) and hindcasted SIMAR (black) time series of Hs, Tp and wave direction. RMSE and correlation are quoted for the wave direction (for Hs and Tp the values are quoted in Figure 6).**

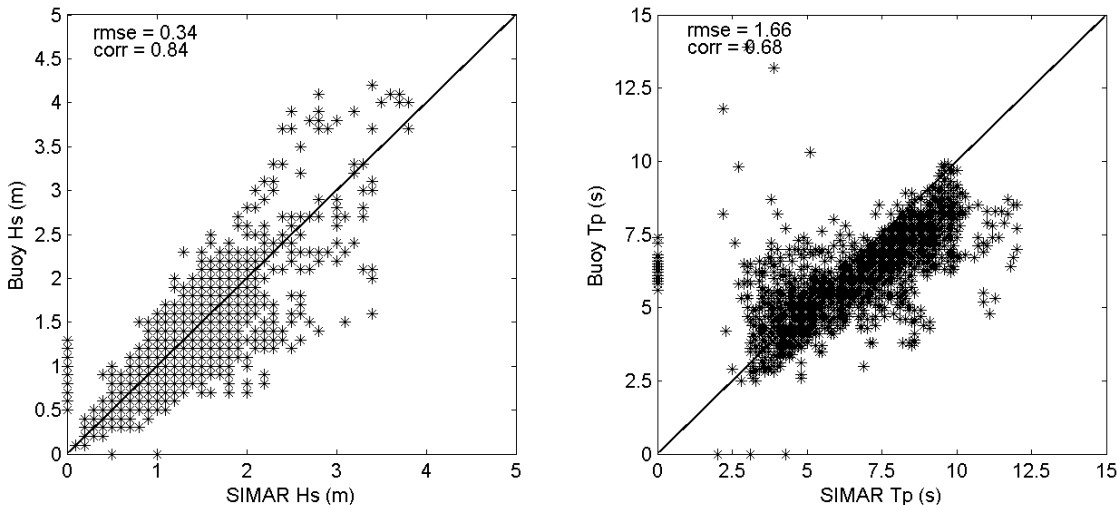

**Figure 6 Scatter plots of buoy observations vs SIMAR hindcast for Hs (left) and Tp (right). RMSE and correlation are quoted in each figure.**





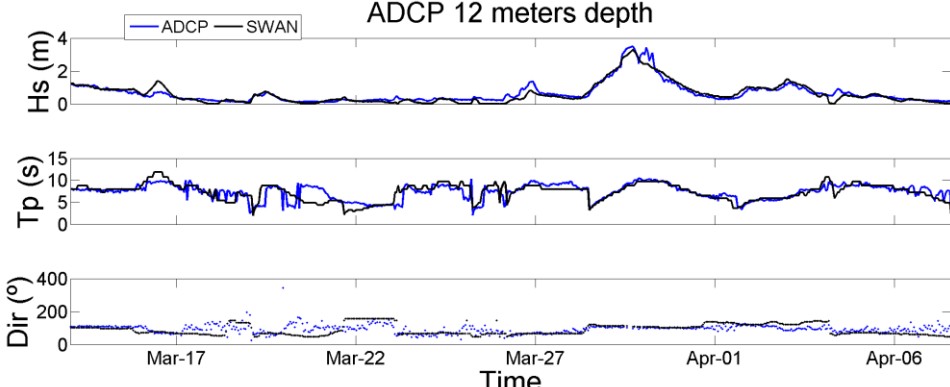

**Figure 7: Hs, Tp and wave direction as modelled by SWAN and observed at the ADCP deployed at 12m depth in Cala Millor beach.**

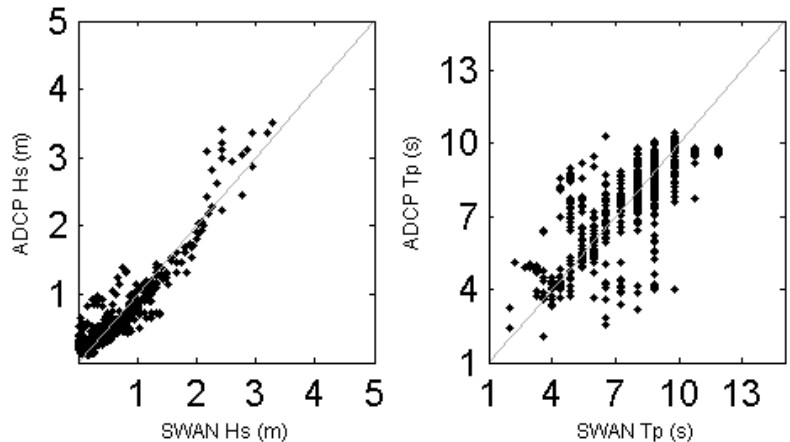

**Figure 8: SWAN vs ADCP scatter plots of Hs (left) and Tp (right) in Cala Millor.**





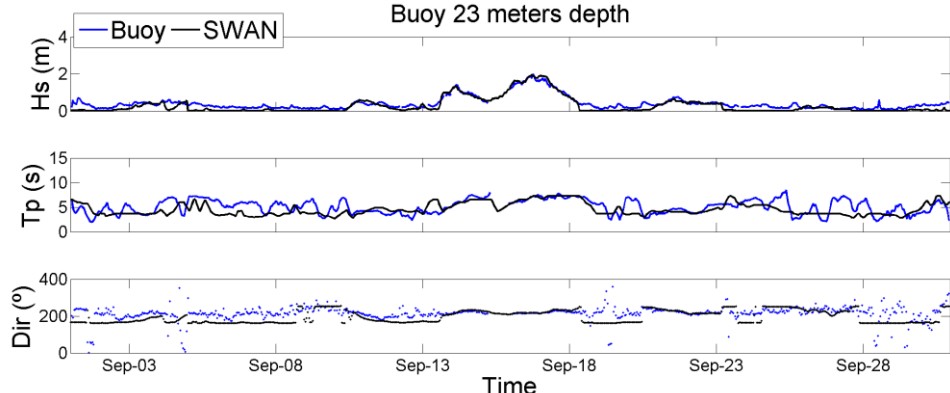

**Figure 9: Hs, Tp and wave direction as modelled by SWAN model and observed at the buoy deployed at 23 m depth in Playa de Palma beach.**

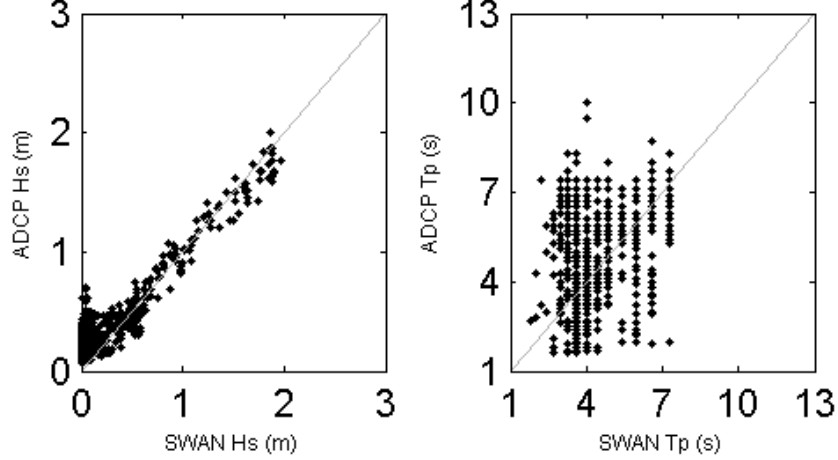

5 **Figure 10: SWAN vs ADCP scatter plots of Hs (left) and Tp (right) in Playa de Palma.**




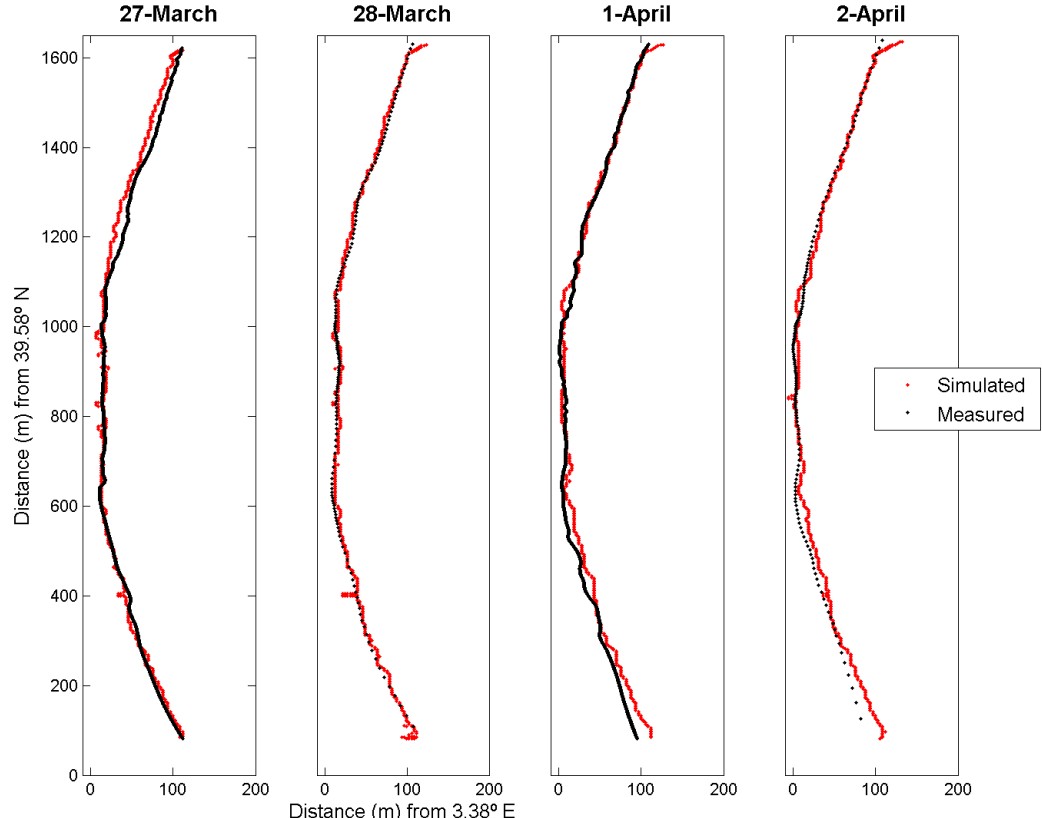

**Figure 11. Observed (black) and modelled by SWASH (red) shoreline positions in Cala Millor**




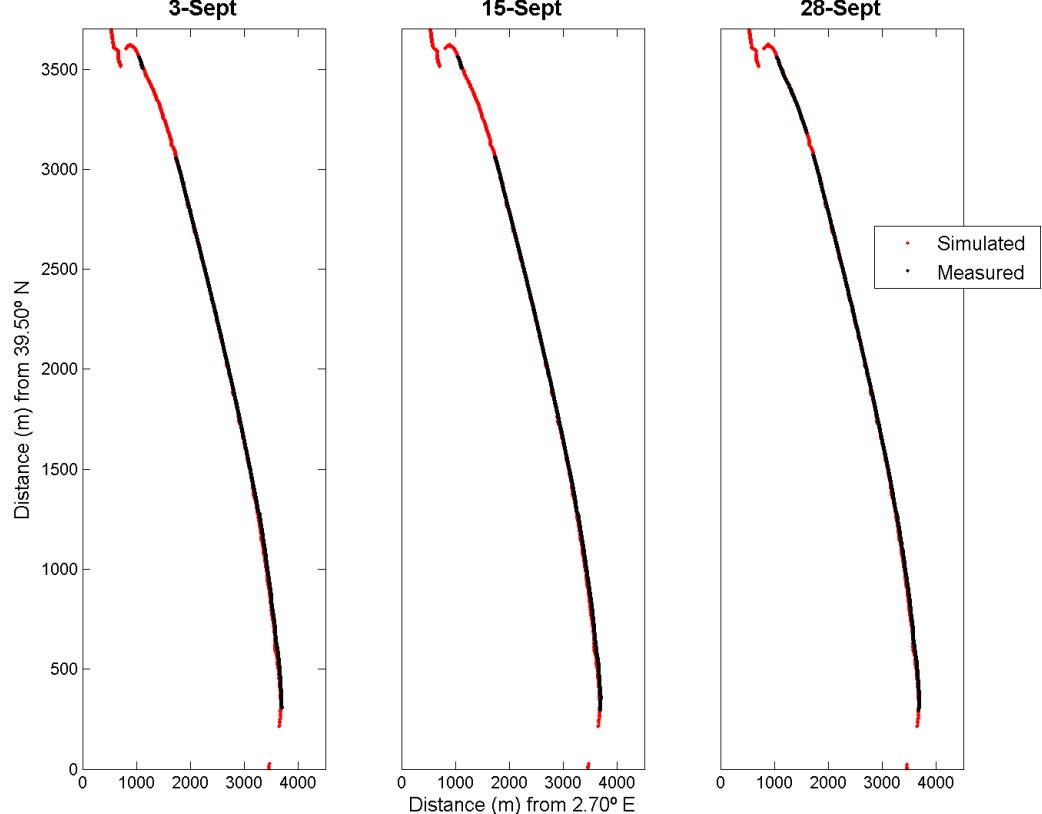

**Figure 12. Observed (black) and modelled by SWASH (red) shoreline positions in Playa de Palma**




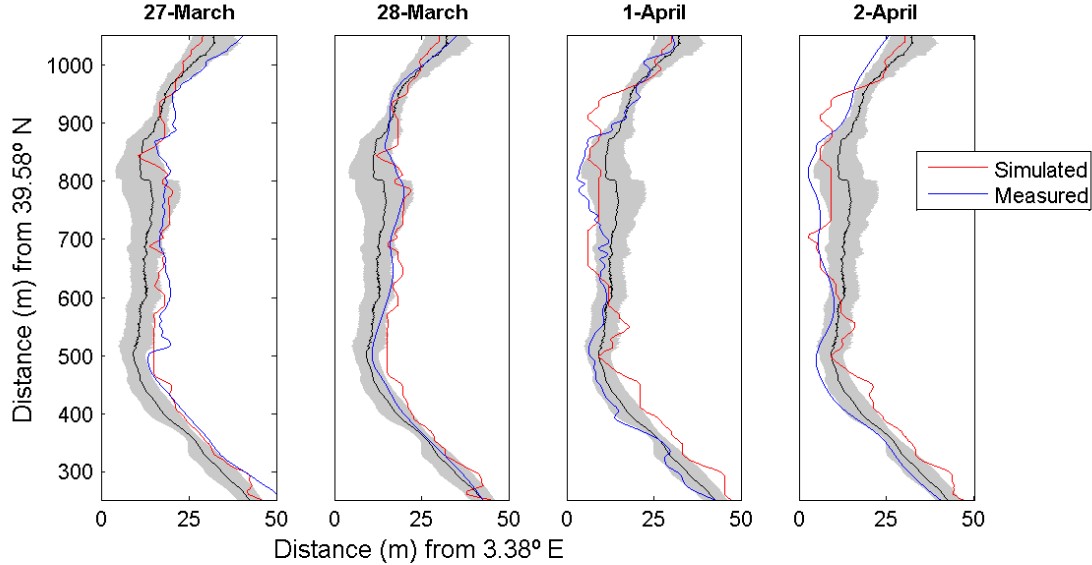

**Figure 13. Modelled (red) and observed (blue) shorelines positions in Cala Millor with mean shoreline position (black line) and its std (grey shadow) zoomed to the central sector.**



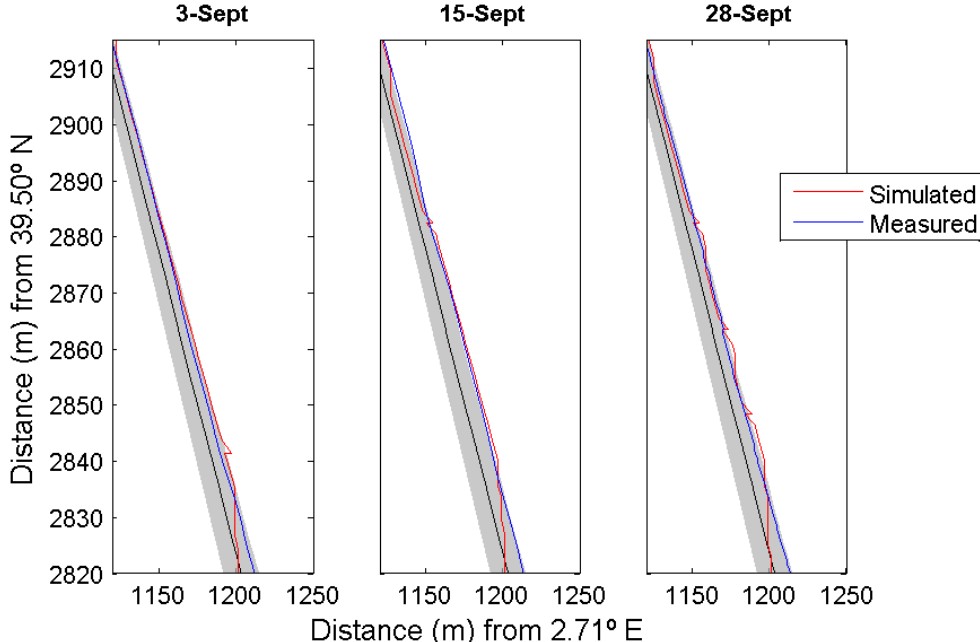

**Figure 14. Modelled (red) and observed (blue) shorelines positions in Playa de Palma with mean shoreline position (black line) and its std (grey shadow).**

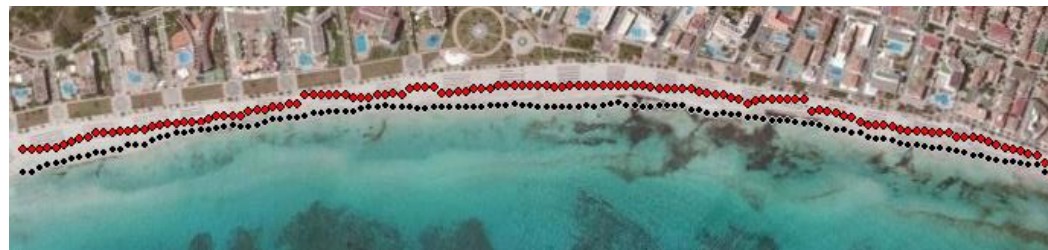

**Figure 15: Present-day shoreline position in (black) and landward migration (in red) in the worst case scenario (mean sea level rise under RCP85 and extreme wave conditions) by the end of the 21st century in Cala Millor beach.**

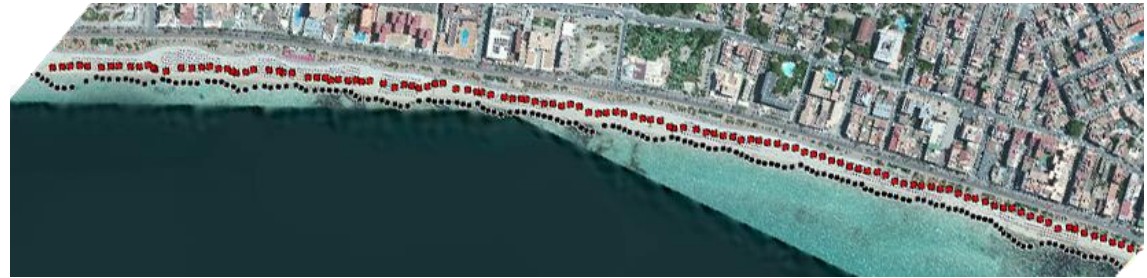

**Figure 16. Present-day shoreline position in (black) and landward migration (in red) in the worst case scenario (mean sea level rise under RCP85 and extreme wave conditions) by the end of the 21st century in Playa de Palma beach.**