# Peer review of "Changes in beach shoreline due to sea level rise and waves under climate change scenarios: application to the Balearic Islands (Western Mediterranean)."

_Natural Hazards and Earth System Sciences, 2016_

## Referee Comment (RC1) · Anonymous Referee #1 · 8 Jan 2017

The paper presents the contribution of sea level rise and waves to shoreline changes under different scenarios. It is an interesting topic that in the last decades has gained much attention and the paper is within the remit of the journal; however, I suggest some reworking before publication as some sections are unclear as well as some figures and results missing. I really hope that the following comments will improve the quality of the paper.

Introduction: The authors introduce properly the main topic in the first three paragraphs; however, from line 15 to line 24 on page 2 the author focuses on the methodology and the used data, which is described on Data and Methods section. I would recommend to remove it in order to avoid repetitions.

Data and Methods: Information about the study area which is provided in the conclusions section ( i.e. bed rock and small cliff at Calla Millor, line 24 on Page 9) should be displaced in the description of the studied areas.

Video monitoring data: the description of the SIRENA system should include information about the errors of intrinsic and extrinsic parameters. Moreover, the error of the observed shoreline position should have been displayed in Figures 11 and 12.

Forcing of numerical models: The wave climate analysis is unclear and results of wave projections are not shown in figures. Authors should provide the projections obtained from WAM model as well as the mean and extreme wave changes over the period from 2080 and 2100 and from 1980-2000. Moreover, a figure of the significant wave height versus return period should be included so that the reader can have a perception of the fit.

Shoreline changes under climate change scenarios:The authors assume that the beach profile remains constant and although they apply PETRA model in order to check the limitation of the assumption, they do not show any figures or results.

Other comments:

Line 11 (Page 7):In Playa de Palma beach, differences are shown between measured and modeled peak period. A reason would be needed here.

Line28 (Page 7): It is said that the difference between measured and simulated shoreline is related to the distance of the cameras. How much is the error of the measured shoreline? This should be indicated.

Lines 29-31 (Page 9)- Line 1 (Page 10): It is justified that the beach shape is constant due to the low variability in the energy flux and that the beach profile plays a minor role in the shoreline retreat. Nevertheless, neither results nor figures are provided in the

manuscript. In my view, they should have been provided.

Language issues:

The paper uses multiple verb forms. For example, on line 8 (Page 4) present in the passive voice is used ('models are combined') and on line 15 (Page 4) past in the same voice is selected ('simulations were performed').Perhaps a revision of the manuscript should be needed.

Please avoid the use of first personal along the manuscript. As you probably know, scientific studies do not recommend it.

References:

References are not properly written.On the reference list, the initials of the author(s) must be written always after the last name and the manuscript does not supply the correct order. I suggest reading carefully manuscript preparation guidelines. Finally, another mistake that I would like to mention:

The in-text citation of Vera Guimarães et al. (2015) is displayed as Guimarães on the reference list.
* * *

---

## Referee Comment (RC2) · Anonymous Referee #2 · 20 Jan 2017

This study investigates hydrodynamic conditions for two beaches in the Balearic islands and estimates the evolution of a shoreline proxy under different sea-level rise and waves conditions. The hypothesis of the study are clearly exposed and the topic is important. In particular, the authors attempt to assess the impacts of sea-level rise and changing waves conditions, which are usually considered negligeable in many studies. I think it is relevant for NHESS and could be published with moderate revisions.

Moderate comments:

The study is first of all an hydrodynamic study:  it assumes that the nearshore

bathymetry is unchanged over multi-decadal timescales, and estimates how the position of an hydrodynamic shoreline proxy evolves with changing offshore conditions. The authors provide a justification to this modeling strategy in their conclusion. However, in practice, assuming no change in the beach morphology as sea-level rises appears as a very optimistic assumption. Some references in the manuscript support this statement,. Other suggested references: e.g. Ranasinghe et al. 2012, Climatic Change; Davidson Arnott et al., 2002, Journal of Coastal Research.

To justify their modeling choice the authors could say that they assess a minimum impact to be expected from sea-level rise and changing waves conditions, with the assumption that the sedimentary budget over these two beaches remains in equilibrium. Assessing minimum impacts of sea-level rise is useful for decision makers as it defines the minimum adaptation needs. Data regarding the evolution of these beaches over the last decades would be useful for the reader to understand this sedimentary context. Do the authors have access to historical aerial photographs that would allow to appreciate how the two sites have evolved over these timescales?

This study would also benefit from better explainations regarding the uncertainties: in particular, there are some confusions regarding the +/- 1 /Sigma uncertainties arround median sea-level rise values for scenarios RCP 4.5 and 8.5. The authors interpret them as minima / maxima, which is not true, as there are difficulties in defining boundaries in future sea-level rise (so-called high-end and low-end scenarios). I suggest to revise this aspect, especially page 8 (line 20: RCP85 +1\Sigma is not a worst case scenario).

Some detailled comments follow. I hope these comments are useful.

Details: -Abstract: please note that the coastline generally refers to a marker such as the dune toe. Here, the authors investigate changes in a shoreline proxy: the limit of the swash. I suggest to precise in the manuscript which shoreline proxy has been choosen and why it is relevant in the context of Mallorca. I guess that the beach width is especially important for tourism (?). To support this discussion, Boak and Turner

2004 (Journal of Coastal Research) would be an approrpiate reference.

Line 30 page 9: "The justification of constant beach shape is largely justified by the fact that there are no significant changes in the energy flux in wave projections that may force a change in the shape due to oceanic forcing." I strongly advise to revise this sentence: with higher sea-levels and identical hydrodynamic conditions, the beach profile is expected to translate or change. I advise the authors discuss the litterature dedicated to beaches equilibrium profiles and beaches morphodynamics. Another paper that could be useful would be Stive et al 2002.

Figure 12: the scale of this figure is not adequate given the scale of the processes to be observed. I suggest to redo this figure

I suggest to avoid abbrevations when not necessary (e.g. std in the legend of figure 13)

Suggested references Boak, E. H., & Turner, I. L. (2005). Shoreline definition and detection: a review. Journal of coastal research, 688-703. Davidson-Arnott, R. G. (2005). Conceptual model of the effects of sea level rise on sandy coasts. Journal of Coastal Research, 1166-1172. Ranasinghe, R., Callaghan, D., & Stive, M. J. (2012). Estimating coastal recession due to sea level rise: beyond the Bruun rule. Climatic Change, 110(3), 561-574. Stive, M. J., Aarninkhof, S. G., Hamm, L., Hanson, H., Larson, M., Wijnberg, K. M., ... & Capobianco, M. (2002). Variability of shore and shoreline evolution. Coastal Engineering, 47(2), 211-235.

---

## Author Comment (AC1) · 21 Feb 2017

**Reviewer #1:**

**The paper presents the contribution of sea level rise and waves to shoreline changes under different scenarios. It is an interesting topic that in the last decades has gained much attention and the paper is within the remit of the journal; however, I suggest some reworking before publication as some sections are unclear as well as some figures and results missing. I really hope that the following comments will improve the quality of the paper.**

We deeply thank the referee's comments and the effort he/she made in reviewing carefully our work. In the new version of the manuscript we have implemented all the points raised in the review. We think that, thanks to this discussion, the new version of the manuscript has been improved.

1. **Introduction: The authors introduce properly the main topic in the first three paragraphs; however, from line 15 to line 24 on page 2 the author focuses on the methodology and the used data, which is described on Data and Methods section. I would recommend to remove it in order to avoid repetitions.**

   We have reworded this paragraph. Now it is much shorter and simply mentions the combined use of sea level and wave projections, leaving the details for section 2. We have also moved lines 25-26 to the Data and Methods section (line 2 in page 3) in order to avoid repetitions and to improve the readability of the paper.

2. **Data and Methods: Information about the study area which is provided in the conclusions section (i.e. bed rock and small cliff at Calla Millor, line 24 on Page 9) should be displaced in the description of the studied areas.**

   We have moved this description to the beginning of the Data and Methods section, which now reads as:

   *"Cala Millor and Playa de Palma are two micro-tidal sandy beaches located in Mallorca Island (Balearic Islands, Western Mediterranean Sea, Figure 1). Cala Millor is 1.7 Km along-shore by 35-40 m cross-shore, with a bed rock and a small cliff at the southernmost sector of the beach, and it is exposed to offshore waves from NE to ESE."*

3.  **Video monitoring data: the description of the SIRENA system should include information about the errors of intrinsic and extrinsic parameters. Moreover, the error of the observed shoreline position should have been displayed in Figures 11 and 12.**

    Section 2.3 describing the video monitoring data has been extended to incorporate the information required by the reviewer, including the references to the intrinsic and extrinsic errors. We have also modified Figure 15 to display the error of the observed shoreline position.

4.  **Forcing of numerical models: The wave climate analysis is unclear and results of wave projections are not shown in figures. Authors should provide the projections obtained from WAM model as well as the mean and extreme wave changes over the period from 2080 and 2100 and from 1980-2000. Moreover, a figure of the significant wave height versus return period should be included so that the reader can have a perception of the fit.**

    We have improved section 2.5 (on the forcing of the numerical models) rewriting part of the section and adding details where we considered there could be confusion. We have also produced the figure suggested by the reviewer (new figure 5) showing the evolution of Hs in one of the wave projections for the mean and extreme regimes.

[Figure]

Figure 5: Return periods in A2 scenario for future projections (blue dashed line), control simulation (black dotted line) and hindcast (red line). Note that there are different time periods for the series as well as the overlapping of hindcast and control scenario. The red line indicates the first day of hindcast time series.

5. **Shoreline changes under climate change scenarios: The authors assume that the beach profile remains constant and although they apply PETRA model in order to check the limitation of the assumption, they do not show any figures or results.**

We have now extended the results from PETRA and included a new figure with the outputs of this model that justify our assumption. The new figure 16 represents the changes in the central cross-shore profile under present-day conditions and under sea level rises of 0.5m and 0.9m. Results indicate a change of less than 20 cm. This is now discussed in the text (section 4, 2nd paragraph).

[Figure]

**Figure 16: Changes in the cross-profile in Cala Millor (left panel) and Playa de Palma (right panel) in nearshore area under different sea levels.**

6. **Line 11 (Page 7): In Playa de Palma beach, differences are shown between measured and modeled peak period. A reason would be needed here.**

We have added a comment on the differences found for Tp. To our understanding these may arise from i) noise in measurements and ii) local wind variability (the model just propagates the waves from the boundary but does not force them with local winds). This is now stated in the text (section 3.1, last paragraph).

7. **Line28 (Page 7): It is said that the difference between measured and simulated shoreline is related to the distance of the cameras. How much is the error of the measured shoreline? This should be indicated.**

We have added here a reference to section 2.3, where the errors of the cameras are now explained in detail.

8. **Lines 29-31 (Page 9)- Line 1 (Page 10): It is justified that the beach shape is constant due to the low variability in the energy flux and that the beach profile plays a minor role in the shoreline retreat. Nevertheless, neither results nor figures are provided in the manuscript. In my view, they should have been provided.**

We have modified the paragraph in section 4 referring to the beach shape and also the summary section. We have assumed constant beach shape for a number of reasons: i) our numerical models do not reproduce the morphological response to the beach shape in the long term, ii) the shape is largely controlled by anthropogenic activities on the emerged beach, to increase to comfortability of tourism, iii) we assume that the effects of increasing mean sea level is much larger than any sediment redistribution and iv) at least for Cala Millor, the projected changes in mean wave direction are smaller than the natural variability..

9. **Language issues:**
**The paper uses multiple verb forms. For example, on line 8 (Page 4) present in the passive voice is used ('models are combined') and on line 15 (Page 4) past in the same voice is selected ('simulations were performed').Perhaps a revision of the manuscript should be needed. Please avoid the use of first personal along the manuscript. As you probably know, scientific studies do not recommend it.**

We are grateful for the suggestion, the manuscript has been revised and it had been tried to improve the language.

10. **References:**
**References are not properly written.On the reference list, the initials of the author(s) must be written always after the last name and the manuscript does not supply the correct**

**order. I suggest reading carefully manuscript preparation guidelines. Finally, another mistake that I would like to mention: The in-text citation of Vera Guimarães et al. (2015) is displayed as Guimarães on the reference list.**

We have revised and corrected the references to homogenise the format.

[revised manuscript text omitted]

---

## Author Comment (AC2) · 21 Feb 2017

**Reviewer #2:**

**This study investigates hydrodynamic conditions for two beaches in the Balearic islands and estimates the evolution of a shoreline proxy under different sea-level rise and waves conditions. The hypothesis of the study are clearly exposed and the topic is important. In particular, the authors attempt to assess the impacts of sea-level rise and changing waves conditions, which are usually considered negligible in many studies. I think it is relevant for NHESS and could be published with moderate revisions.**

We are grateful to the referee for the constructive comments provided. We have followed all his/her suggestions, which we believe have helped to improve our manuscript.

1. **The study is first of all a hydrodynamic study: it assumes that the nearshore bathymetry is unchanged over multi-decadal timescales, and estimates how the position of an hydrodynamic shoreline proxy evolves with changing offshore conditions. The authors provide a justification to this modeling strategy in their conclusion. However, in practice, assuming no change in the beach morphology as sea-level rises appears as a very optimistic assumption. Some references in the manuscript support this statement. Other suggested references: e.g. Ranasinghe et al. 2012, Climatic Change; Davidson Arnott et al., 2002, Journal of Coastal Research.**

   We agree with the reviewer and we are aware of the limitations of our approach. Following also the other reviewer's comments we have extended the discussion on the assumptions of constant beach profile and shape. We have also made further tests using the PETRA model to demonstrate the appropriateness of these assumptions in the context of our study.

   Regarding the references provided by the reviewer, they refer to natural environments in which the shoreline may migrate landward in response to sea level rise. This is not the case in our study, where sediment supply does not exist due to the anthropization of the beaches. Nevertheless, we appreciate the examples; they have been referred to in the conclusions as cases in which the beach morphology changes.

2. **To justify their modelling choice the authors could say that they assess a minimum impact to be expected from sea-level rise and changing waves conditions, with the assumption that the sedimentary budget over these two beaches remains in equilibrium. Assessing minimum impacts of sea-level rise is useful for decision makers as it defines the minimum adaptation needs.**

We agree; this has been stated in the introduction and conclusion sections.

3. **Data regarding the evolution of these beaches over the last decades would be useful for the reader to understand this sedimentary context. Do the authors have access to historical aerial photographs that would allow to appreciate how the two sites have evolved over these timescales?**

Since the 1970s many coastal zones in Mallorca have undergone a strong urbanization, with natural environments such as dunes becoming heavily exploited touristic resources. This has made that during the last 50 years the sedimentary budget in the areas of our study have been far from being natural. There are aerial photographs of the evolution of the two zones, but what they reflect is the urban development rather than the natural coastal evolution of the beaches.

4. **This study would also benefit from better explanations regarding the uncertainties: in particular, there are some confusions regarding the +/- 1 /Sigma uncertainties around median sea-level rise values for scenarios RCP 4.5 and 8.5. The authors interpret them as minima / maxima, which is not true, as there are difficulties in defining boundaries in future sea-level rise (so-called high-end and low-end scenarios). I suggest to revise this aspect, especially page 8 (line 20: RCP85 +1\Sigma is not a worst case scenario).**

We apologise for the confusing wording. Indeed, what we called "worst case scenario" is not the so-called high-end scenario. Therefore, the terms have been reworded. As stated in the text, the uncertainties refer to the ensemble model dispersion. So we now refer simply to "RCP85 plus 1-sigma" or the "upper uncertainty limit".

5. **Details: -Abstract: please note that the coastline generally refers to a marker such as the dune toe. Here, the authors investigate changes in a shoreline proxy: the limit of the swash. I suggest to precise in the manuscript which shoreline proxy has been chosen and why it is relevant in the context of Mallorca. I guess that the beach width is especially important for tourism (?). To support this discussion, Boak and Turner 2004 (Journal of Coastal Research) would be an appropriate reference.**

We have changed the abstract to account for the reviewer's comment. We have also explicitly stated in the introduction that our focus is on the shoreline position, to avoid confusion with the widely used term of coastline (section 1, 3rd paragraph). In addition, we also mention that changes in the shoreline are a strong impact for the tourism-oriented beaches (same paragraph).

6. **Line 30 page 9: "The justification of constant beach shape is largely justified by the fact that there are no significant changes in the energy flux in wave projections that may force a change in the shape due to oceanic forcing." I strongly advise to revise this sentence: with higher sea-levels and identical hydrodynamic conditions, the beach profile is expected to translate or change. I advise the authors discuss the literature dedicated to beaches equilibrium profiles and beaches morphodynamics. Another paper that could be useful would be Stive et al 2002.**

We have rewritten most of section 4 and tried to better explain our approach. We agree with the reviewer and we are aware of the limitations of our modelling. Further analyses have been done with respect to the assumption of unchanged beach profile and shape (see the response to reviewer 1 for more details); they are now discussed with more depth in section 4. We also clearly state that our results are a lower boundary of the expected shoreline retreat, as we are neglecting other processes such as erosion.

7. **Figure 12: the scale of this figure is not adequate given the scale of the processes to be observed. I suggest to redo this figure.**

We understand the point made by the reviewer. Nevertheless we would prefer to keep these figures as: 1) they provide an overview of the overall performance of the models along the entire length of the two beaches; 2) the changes are clear in Figures 12 and 13, which are zoomed versions of the former.

8. **I suggest to avoid abbreviations when not necessary (e.g. std in the legend of figure 13)**

We have removed these abbreviations.

9. **Suggested references Boak, E. H., & Turner, I. L. (2005). Shoreline definition and detection: a review. Journal of coastal research, 688-703. Davidson-Arnott, R. G. (2005). Conceptual model of the effects of sea level rise on sandy coasts. Journal of Coastal Research, 1166-1172. Ranasinghe, R., Callaghan, D., & Stive, M. J. (2012). Estimating coastal recession due to sea level rise: beyond the Bruun rule. Climatic Change, 110(3), 561-574. Stive, M. J., Aarninkhof, S. G., Hamm, L., Hanson, H., Larson, M., Wijnberg, K. M., ... & Capobianco, M. (2002). Variability of shore and shoreline evolution. Coastal Engineering, 47(2), 211-235.**

We are thankful to the referee for suggesting these very interesting references.

[revised manuscript text omitted]

---

## Referee Report (RR1)

This paper examines the impacts of future sea level rise and a changing wave climate on shoreline positions of two beaches in the western Mediterranean. The authors' analysis is sound; however some details in the methodology were omitted and need to be clarified. This is an important topic that fits the scope of NHESS and I suggest publication after moderate revisions.

1.  Section 1: Additional papers that could be referenced:

    Passeri et al., 2015, doi: 10.1002/2015EF000298
    Gutierrez et al., 2011, doi 10.1029/2010JF001891
    Plant et al., 2016, doi: 10.1002/2015EF000331

    While the authors are neglecting coastal erosion in their projections, I think it is important for them to mention that this study goes beyond "bathtub" approximations of sea level rise (see Passeri et al., 2015) – a bathtub approach would simply assume that future coastal retreat would be at the 1 m contour for 1 m rise in sea level. Rather, the authors are dynamically simulating waves and water levels under a changing climate and SLR to determine the future shoreline position. This provides additional novelty for the paper.

2.  Section 2.4: What is the overland extent of the SWASH model? Does SWASH resolve wave runup? This would be necessary to accurately compare the wet-dry shoreline with the video footage. Since you are looking at the wet-dry interface as proxy for the shoreline, how does SWASH resolve wetting and drying processes? This should be mentioned in the model description.

3.  Section 2.5: Is the SWASH model forced with tides? How is sea level rise incorporated into the models?

4.  Section 4: A brief description of the PETRA model should be included. I would also consider moving this to the methodology since it is what you are basing your assumption of a constant beach profile on.

5.  Section 4: This is the first mention of the wall backing the beach. A better description of the study area is needed at the beginning of the manuscript. Also, do the beaches have dunes/what is the elevation of the dune or berm? Dune height has been linked to long-term shoreline change (see Plant et al., 2016) and would affect the inundation extent. The authors conclude that coastal retreat is lower in Playa de Palma due to a steeper beach slope – again, this is the first mention of the beach slope. By moving the discussion of the PETRA model to the methodology, this would help to better describe the study area. Lastly, are these beaches nourished? If so, this could help to justify neglecting coastal erosion.

Minor edits: I suggest the authors review the paper carefully for grammatical errors.

Page 2 Line 32: Should be "SOM graphically display".

Page 3 Line 27: Define IMEDEA acronym.

Page 3 Line 28: Morphodynamics is misspelled.

Page 7 Line 1: I think this is incorrectly referenced to Figure 7 when it should be Figure 6.

Page 7 Line 3: Since your plot is in degrees, you should define what N and SE are in degrees to make it easier for the reader.

Page 7 Line 16: Should be "over 0.9".

Page 7 Line 24: Use the exact number for the correlation coefficient rather than rounding to 0.3 – should match the correlation coefficient in your table.

---

## Author Response (AR2)

**Referee 1**
I thank the authors for considering the questions raised during the review. I think the paper is relevant to NHESS and could be published, provided the article becomes more cautious on the two points bellow:

We are grateful to the reviewer for the comments on our work.

- long term coastal dynamics: many coastal morphodynamics scientists will disagree with the sentence below: "It has have also been demonstrated that the changes in the beach profile play a minor role in the shoreline retreat due to sea level rise and waves. On top of the above reasons, we can hardly avoid the simplifications can be hardly avoided, as numerical approaches reproducing the long term morphological response of the beach do not exist so far." The papers quoted by the authors in the previous sentence and earlier in the manuscript clearly demonstrate that we don't know so much about beach morphodynamics, but that some modeling approaches exist (e.g. Ranasingh et al 2012). I think that at least this sentence should be reconsidered.

We have modified the sentence according to this comment. We have also quoted the new reference provided in the context as indicated

- language on uncertainties: saying that the +/-1-\sigma uncertainty is an "upper uncertainty limit" is not appropriate: the IPCC sea-level report (Ch13) and its supplementary material provide the adequate language to be used in this case
I hope this review is useful.

We agree. This was specified in the text but we have now added a new sentence to highlight that these uncertainties quoted here correspond to ±1-σ (page 8, lines 27-28).

**Referee 2**
The authors have revised the manuscript according to the suggestions of both referees. In this revised version, they have made efforts for improving the quality; however, some concepts are still unclear and in some sections the use of technical and English language should be revised.

We have now followed new reviewers' comments and the manuscript has been further modified. We hope that the new version will be satisfactory for reviewer 2 as well.

**Referee 3**
This paper examines the impacts of future sea level rise and a changing wave climate on shoreline positions of two beaches in the western Mediterranean. The authors' analysis is sound; however some details in the methodology were omitted and need to be clarified. This is an important topic that fits the scope of NHESS and I suggest publication after moderate revisions.

We thank the referee for in depth her/his review.

**1. Section 1: Additional papers that could be referenced:**

Passeri et al., 2015, doi: 10.1002/2015EF000298
Gutierrez et al., 2011, doi 10.1029/2010JF001891
Plant et al., 2016, doi: 10.1002/2015EF000331

While the authors are neglecting coastal erosion in their projections, I think it is important for them to mention that this study goes beyond "bathtub" approximations of sea level rise (see Passeri et al., 2015) – a bathtub approach would simply assume that future coastal retreat would be at the 1 m contour for 1 m rise in sea level. Rather, the authors are dynamically simulating waves and water levels under a changing climate and SLR to determine the future shoreline position. This provides additional novelty for the paper.

We thank the reviewer for pointing this out. We have added a specific comment on this respect together with the references provided.

**2. Section 2.4: What is the overland extent of the SWASH model? Does SWASH resolve wave runup? This would be necessary to accurately compare the wet-dry shoreline with the video footage. Since you are looking at the wet-dry interface as proxy for the shoreline, how does SWASH resolve wetting and drying processes? This should be mentioned in the model description.**

We have extended the model description in section 2.4 to address the issues raised by the reviewer.

**3. Section 2.5: Is the SWASH model forced with tides? How is sea level rise incorporated into the models?**

Regarding the tides, these are very small in the Mediterranean Sea so we have neglected their impact in our assessment. Also, since we are not working in a wide continental shelf, tidal changes are not foreseen as a result of sea level rise.
Regarding the why in which sea level rise has been incorporated, we have done so by including the corresponding still water levels into the model runs corresponding to the selected range of sea level rise projections. This information has been included in section 2.5 for clarification.

**4. Section 4: A brief description of the PETRA model should be included. I would also consider moving this to the methodology since it is what you are basing your assumption of a constant beach profile on.**

We have followed reviewer's advice and have now incorporated the description of the PETRA model in section 2.4 of the methodology.

**5. Section 4: This is the first mention of the wall backing the beach. A better description of the study area is needed at the beginning of the manuscript. Also, do the beaches have dunes/what is the elevation of the dune or berm? Dune height has been linked to long- term shoreline change (see Plant et al., 2016) and would affect the inundation extent. The authors conclude that coastal retreat is lower in Playa de Palma due to a steeper beach slope – again, this is the first mention of the beach slope. By moving the discussion of the PETRA model to the methodology, this would help to better describe the study area. Lastly, are these beaches nourished? If so, this could help to justify neglecting coastal erosion.**

We have extended the description of the beaches in the beginning of the Data and Methods section, including the reference to the steeper slope of Playa de Palma and the fact that both are urban beaches backed by walls and promenades (thus without presence of natural environments such as dunes).
Regarding the beach nourishment, unfortunately there is no data available to be discussed. We found out that both beaches were actually nourished in the 1980s but we have no further information about it.